

# Continued warming of deep waters in Fram Strait

Salar Karam[1], Céline Heuzé[1], Mario Hoppmann[2], and Laura de Steur[3]

[1]Department of Earth Sciences, University of Gothenburg, Gothenburg, Sweden
[2]Alfred-Wegener-Institut Helmholtz-Zentrum für Polar- und Meeresforschung, Am Handelshafen 12, 27570 Bremerhaven, Germany
[3]Norwegian Polar Institute, Tromsø, Norway

**Correspondence:** Salar Karam (salar.karam@gu.se)

**Abstract.** Fram Strait is the only deep gateway between the Arctic and the rest of the World Ocean, and is thus a key region to understand how the deep Arctic will evolve. However, studies and data regarding the deep ocean are scarce, making it difficult to understand its role in the climate system. Here we analyse oceanographic data from the deep ocean from two long-term mooring locations (F11 and HG-FEVI) in Fram Strait between 2010-2023, to investigate long-term changes in the hydrographic properties. We compile hydrographic profile data since the 1980's for additional context in the upstream basins: the Greenland Sea, and the Eurasian Basin. At mooring F11 in western Fram Strait, we find a clear seasonality, with increased Greenland Sea Deep Water (GSDW) presence during summer, and increased Eurasian Basin Deep Water (EBDW) presence during winter. Evaluating long-term changes, we find a modest temperature increase of ∼ 0.1 °C for EBDW since the 1980's. For GSDW, south of Fram Strait, we find a strong temperature increase of ∼ 0.4-0.5 °C for the same period. The different warming rates have led to GSDW becoming warmer than EBDW since ∼ 2017/2018. This means that the Greenland Sea is no longer a heat sink for the Arctic Ocean at depth, but a heat source. It is therefore possible that EBDW temperatures will increase faster in the future.

## 1 Introduction

The Arctic Ocean has drastically transformed in recent decades, which has not been limited only to the surface waters, but has also extended to the deep with a warming of intermediate and deep waters (e.g. Somavilla et al., 2013; von Appen et al., 2015; Lauvset et al., 2018; Abot et al., 2023). Dense intermediate and deep waters from the Greenland Sea and the central Arctic Ocean supply much of the deep waters of the Atlantic Ocean (e.g. Tanhua et al., 2005; Brakstad et al., 2023), which are instrumental for the meridional overturning circulation (e.g. Tsubouchi et al., 2021). Since the halt of deep convection in the Greenland Sea in the 1980's (e.g. Bönisch et al., 1997; Budeus et al., 1998), the lack of formation of the cold and fresh Greenland Sea Deep Water (GSDW) has been compensated by an increased inflow of relatively warm and salty Eurasian Basin Deep Water (EBDW; Somavilla et al., 2013). This increased inflow has led to a rapid warming of GSDW with a warming trend



of 0.11 °C decade$^{-1}$ between 1997-2014 (e.g. Somavilla et al., 2013; von Appen et al., 2015), an order of magnitude higher than in the rest of the deep World Ocean (e.g. Desbruyères et al., 2016). During the same period, EBDW has also warmed

at a rate of 0.05 °C decade$^{-1}$ (e.g. von Appen et al., 2015). This warming has direct implications for sea level rise due to thermal expansion (e.g. Fasullo and Gent, 2017). Considering a rapidly changing Arctic, and indeed global climate system, it is becoming increasingly important to investigate the role of changes in the deep Arctic and its connection to the rest of the World's ocean.

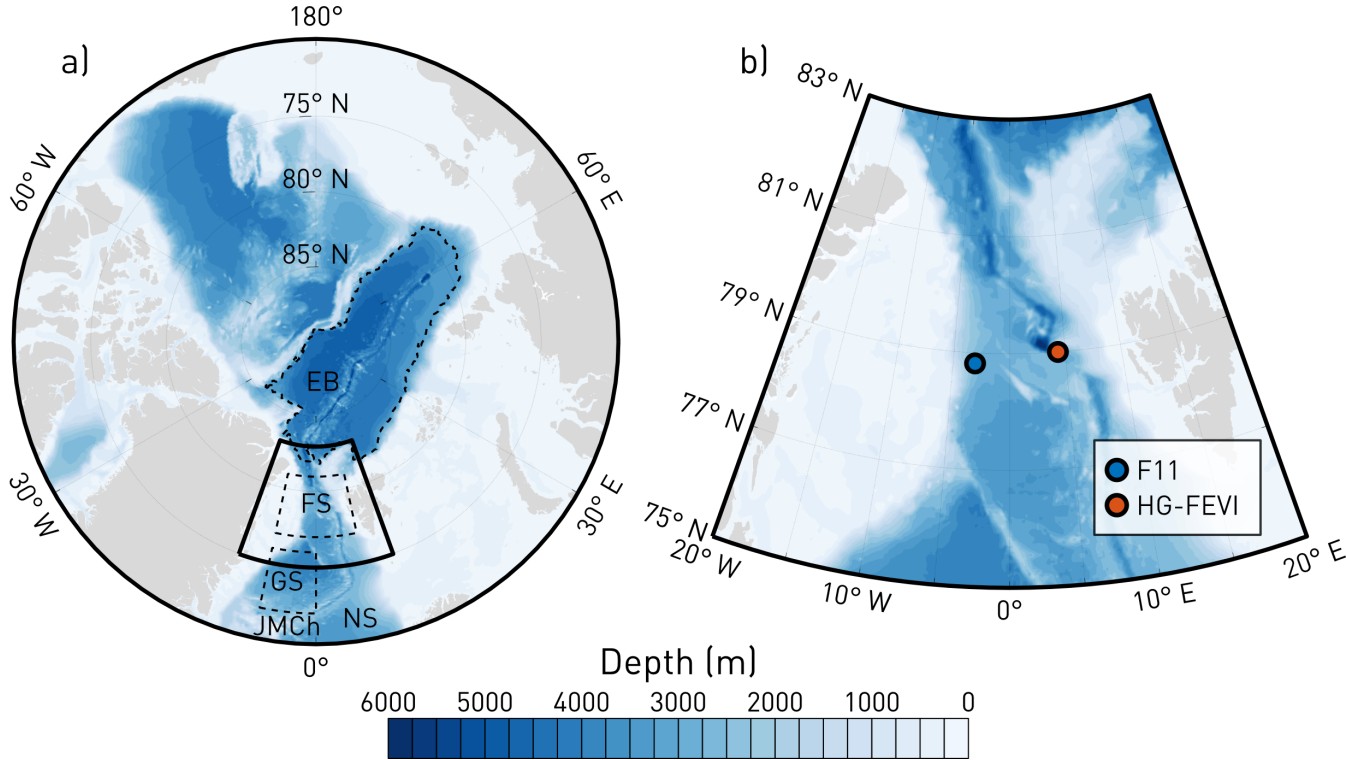

**Figure 1.** Map showing the bathymetry of the Arctic Ocean (blue-white colour scale), from the International Bathymetric Chart of the Arctic Ocean (IBCAO; Jakobsson et al., 2020). The grey land mask was obtained from the Global, Self-consistent, Hierarchical, High-resolution Shoreline DataBase (GSHHG; Wessel and Smith, 1996). a) The thin dashed lines indicate the boundaries of the regions from where we compile hydrographic data (see Sec. 2.1). The thick solid line highlights the area zoomed in on the right panel. The locations mentioned in the main text are the Eurasian Basin (EB), Fram Strait (FS), Greenland Sea (GS), Norwegian Sea (NS), and Jan Mayen Channel (JMCh). b) Inset of the left panel, highlighting the Fram Strait area. The coloured dots indicate the locations of the moorings F11 (blue), and HG-FEVI (red).

Situated between Greenland and Svalbard, the approximately 2500-m deep Fram Strait is the only deep gateway connecting

the central Arctic Ocean to the Nordic Seas (Fig. 1). Fram Strait is thus a key region where the Arctic-derived EBDW, formed by shelf-slope convection and entrainment of intermediate waters, mixes with waters from the Nordic Seas, such as GSDW, which was formed by open-ocean convection in the Greenland Sea gyre (e.g. Rudels, 2012; Langehaug and Falck, 2012).




Based on their different warming rates, the colder and fresher, but overall denser GSDW was predicted to reach the same temperature as EBDW in 2020 (Somavilla et al., 2013; von Appen et al., 2015). Associated with these temperature changes,

GSDW has also become increasingly saline, but these changes are not density-compensated (Somavilla et al., 2013). Overall, these changes have led to GSDW now being a lighter water mass than EBDW. If the density gradient between EBDW and GSDW is an important forcing mechanism for driving exchange, it is possible that deep-sea exchange across Fram Strait might qualitatively change.

Knowledge of the variability, trends, and dynamics driving exchange across Fram Strait has increased in recent years thanks

to long-term monitoring by a mooring array across the strait around 78.8-79°N. From 1997 onward, several moorings have been deployed in Fram Strait, jointly maintained by the Alfred Wegener Institute (AWI; Schauer et al., 2004; Beszczynska-Möller et al., 2012; von Appen et al., 2015), and the Norwegian Polar Institute (NPI; de Steur et al., 2014). These studies, however, tend to focus on upper ocean (e.g. Karpouzoglou et al., 2022; de Steur et al., 2023) or Atlantic Water (AW) variability (e.g. Schauer et al., 2004; Beszczynska-Möller et al., 2012; von Appen et al., 2016). Studies of oceanographic conditions in

the deep sea have been relatively scarce, and to our knowledge only a single study pertaining to the mooring array in Fram Strait and the deep ocean has come out (von Appen et al., 2015). Importantly, it was shown that the mean flows were not responsible for advecting deep water masses across the strait, but rather that the deep ocean was driven by surface (barotropic equivalent) mesoscale flows (von Appen et al., 2015). Hydrographic measurements from ships have contributed to elucidating the distribution of deep water masses and long-term changes in Fram Strait (e.g. Rudels et al., 2005; Langehaug and Falck,

2012; Somavilla et al., 2013; Marnela et al., 2016), but have typically been limited to summertime observations. As such, knowledge of the year-round variations and long-term trends in the deep-sea warming in Fram Strait, since 2012/2014, is lacking.

Since 2010, the mooring array in Fram Strait has been equipped with more and better instrumentation (SBE37 MicroCATs), allowing us to distinguish small-scale changes in temperature, as well as to measure salinity at depth (e.g. Karpouzoglou et al.,

2022). To put the mooring data into context, we first compile a large hydrographic profile data set to evaluate the hydrographic property evolution since the 1980's. We then focus on the recent time series of year-round mooring data in Fram Strait, examining the differences between the two mooring sites in eastern and western Fram Strait, and evaluating long-term changes (section 3.1). We do this to determine whether GSDW has warmed to the same temperature as EBDW, as predicted by von Appen et al. (2015), and if so, since when. Furthermore, in section 3.2, we investigate the drivers of these changes. We finish

with a discussion of the larger scale processes that impact the temperature, and the cause of the emergence of seasonality, before concluding with a summary and wider perspective.

## 2   Data and Methods

### 2.1   Hydrographic profiles

To get an overview of the long-term changes of the hydrographic properties in the Arctic, we compiled hydrographic profiles

from multiple sources. We sourced data from the Unified Database for Arctic and Subarctic Hydrography (UDASH; Behrendt





et al., 2018), from the World Ocean Database (WOD18; Boyer et al., 2018), and from Argo (Wong et al., 2020). Yearly zonal sections across Fram Strait since 2011, conducted by NPI are also used (Norwegian Polar Institute, 2010; Dodd et al., 2022h, b, c, a, f, g, d, i). Additionally, we added data from individual cruises KVS2007 (Dodd and Hansen, 2011a), KVS2008 (Dodd and Hansen, 2011b), JCR2018 (Hopkins et al., 2019), MOSAiC (Rabe et al., 2022; Tippenhauer et al., 2023), SAS
(Snoeijs-Leijonmalm, 2022; Heuzé et al., 2022), and AO22 (Dodd et al., 2022e). For the UDASH, WOD, and Argo datasets, we excluded all data that has not been quality flagged as good data.

We separated the hydrographic profiles into three regions: Fram Strait (77-81 °N, 12 °W-12 °E), the Greenland Sea (72-76 °N, 12 °W-12 °E), and the Eurasian Basin (mainly following the 3000-m isobath within the Eurasian Basin of the Arctic; see Fig. 1). Since we are interested in exchanges of deep waters across Fram Strait, which has a sill depth of approximately 2500
m (Jakobsson et al., 2020), and to reduce uncertainty from errant measurements, we use depth-averaged properties between 2400-2600 m. The results are insensitive to exact choice depth levels for the averaging. When discussing GSDW and EBDW, we thus use a property-independent definition, based on regions instead, similar to Somavilla et al. (2013).

For all hydrographic data, we calculate Conservative Temperature ($\Theta$) and Absolute Salinity ($S_A$) using the International Thermodynamic Equations of Seawater (TEOS-10; McDougall and Barker, 2011). Here we note that previous studies often
used Potential Temperature and Practical Salinity (e.g. von Appen et al., 2015; Marnela et al., 2016). For $S_A$ we observe a range of 0.08 g kg$^{-1}$ (35.04 to 35.12 g kg$^{-1}$) across all hydrographic profile data, which is a change of 0.0796 (34.8717 to 34.9513) in Practical Salinity at a pressure level of 2500 dbar in Fram Strait. Similarly, for $\Theta$ we observe a range of 0.6 °C (-1.4 to -0.7 °C), which is a change of 0.6007 °C (-1.4021 to -0.8014 °C) in potential temperature referenced to 2500 dbar. Our results are therefore not strictly comparable when discussing absolute values, however we are more interested in property evolution of the
water masses, thus allowing for a general comparison.

## 2.2 Mooring data

To investigate whether GSDW and EBDW have reached the same temperature, we analyse mooring data between 2010-2022. In this paper we consider only the mooring F11 in western Fram Strait, and the mooring HG-IV-FEVI at AWI Hausgarten site HG-IV (herein only referred to as HG-FEVI) in eastern Fram Strait (for exact positions see Table. 1, Fig. 1). We chose these
moorings since they have the longest and most consistent time series in the deep Fram Strait. Other moorings with data in the deep Fram Strait were mooring F10 and HG-N, in western and northern Fram Strait, respectively. They were discarded as they only have had salinity data since 2017. We also note that the 10-12 year time series from F11 and HG-FEVI each consist of multiple subsequent mooring deployments compiled together into one dataset, which are for brevity referred to as a single mooring in this paper.

For our analyses, we only consider deployments with more accurate Seabird SBE37 MicroCATs measuring temperature, conductivity (from which we derive salinity), and pressure, rather than deployments using Aanderaa Recording Current Meters (RCMs). The RCMs had only an accuracy of ± 0.05 °C for temperature, which is close to or even exceeds the observed variability, and did not measure salinity at all (e.g. von Appen et al., 2015). While the RCMs were perhaps more justly utilised when temperature differences between EBDW and GSDW were relatively large (∼0.3 °C in the 1980's), the temperatures of



**Table 1.** Mooring deployments, longitudinal range, latitudinal range, instrument depth range, and total duration since the first deployment. Not all data is always available during the deployments. Since the moorings were not always deployed at the same location, and the instruments not always deployed at the same depth, we give a range for coordinates and depths.

| Mooring name | Longitude | Latitude | Instrument depth (m) | Combined duration |
|---|---|---|---|---|
| F11 | 3.1° W - 2.97° W | 78.8° N - 78.83° N | 2452-2495 | 9 Sep 2011 – 1 Aug 2021 |
| HG-FEVI | 4.03° E - 4.3337° E | 79° N - 79.062° N | 2511-2565 | 10 Jul 2010 – 9 Jul 2022 |

these two water masses had by the early 2010's warmed within 0.05 °C of each other (Langehaug and Falck, 2012; Somavilla et al., 2013; von Appen et al., 2015). Therefore, we now require more accurate instruments, and other parameters such as salinity, to distinguish between the water masses. In comparison to the RCMs, the SBE37 MicroCATs have an accuracy of $\pm$ 0.002 °C for temperature, and $\pm$ 0.003 mS cm$^{-1}$ for conductivity.

Temperature and salinity data from F11 (de Steur et al., 2021) have typically been acquired at 15 minute intervals. After recovery, data are quality controlled, bad data was excluded, and compared against shipboard CTD data from the instrument deployment depth. When needed, an offset or a drift correction to the salinity data was added to line these up with shipboard CTD values at the start and end of the record. A month-long sliding window was applied to exclude outliers that were 3 standard deviations from the monthly mean in the window. Data were then bin averaged or interpolated to hourly resolution. For HG-FEVI (Bauerfeind et al., 2015a, b, c, d, 2016; Salter et al., 2017; von Appen, 2019a, b; Hoppmann et al., 2023b, 2022, 2023a), temperature and salinity data have typically been acquired at 1-hour intervals. Similar to F11, these data were compared to the few deep shipboard CTD casts available for calibration; no drift correction was necessary, but, when necessary, an offset was added to the salinity to line up the deployments. The data were despiked similarly to F11, and bin averaged or interpolated to hourly resolution.

Deployments with offsets in salinity, that were clearly non-physical and could not be corrected with ship CTD data, were manually offset to have the same mean as the rest of the time series. Any deployments with clear drifts that could not be corrected were removed.

For velocity records in the deep, a mix of RCMs and NORTEK Aquadopps have been deployed, acquiring point measurements of velocity. The data were bin averaged or interpolated to hourly resolution. We applied no further treatment to the velocity records.

## 2.3 Cross correlation

To be able to compare our results to previous studies, we followed a similar procedure to von Appen et al. (2015), where we normalised the daily-averaged temperature and salinity data from the moorings, to distinguish between the two water masses. Since both water masses are changing in temperature and salinity over time, we must calculate how the upper and lower bounds of the time series change over time. We define the the upper and lower bounds by dividing the time series into 6-month long





bins and calculating the 5$^{th}$ and 95$^{th}$ percentiles of each bin, to get estimates of the upper and lower bound of water mass

properties. A linear least squares trend was then fitted to these estimates to give the time-evolution of the envelopes at any

point in time. The normalised variables at any given time is then given by:

$$P_{norm} = \frac{P_{obs} - \frac{1}{2}(P_{upper} + P_{lower})}{\frac{1}{2}(P_{upper} - P_{lower})} \tag{1}$$

where $P_{norm}$ is the normalised variable, $P_{obs}$ is the observed property, and $P_{upper}$ and $P_{lower}$ are the upper and lower bound

estimates.

To investigate the temporal variability of the water mass properties, we cross correlate the daily-averaged normalised variables in a 3-month long sliding window, with a 1-day shift. Since temperature and salinity co-vary, we use the correlation at a lag 0. This creates a time series of the correlation between normalised temperature and salinity, allowing us to identify whether the still fresher GSDW is now warmer than EBDW, as was predicted to happen in 2020 (von Appen et al., 2015). Additionally,

we cross correlate the normalised temperature with the meridional and zonal velocity components using a 3-month sliding window, and testing for lags of up to 10 days. The significance of the correlations is determined using a standard t-test, and only correlations significant at 95% are shown.

## 2.4    Regime shift analysis

In order to detect possible regime shifts and their timing, we applied a sequential algorithm for regime shift detection to

time series of cross correlation between temperature and salinity (Rodionov, 2004). The algorithm does not require a priori hypotheses regarding the timing of the regime shifts. It continually tests newer data points if they are significantly different from the mean value and its variance of the current regime. If a statistically significant change point is found, the following data points are used to validate the confidence of the shift in the mean. This is done through a metric called a regime shift index (RSI). RSI is the cumulative sum of the normalised anomalies from the mean of the new regime. If the RSI is positive

for all data points within a specified cut-off period, it means that a regime shift has occurred (Rodionov, 2004; Rodionov and Overland, 2005). In this study, we used a cut-off period of 5 years, similar to de Steur et al. (2023). To test the sensitivity of the length of the cut-off period, we also tested cut-off periods of 4 years and 6 years, but results were similar (not shown).

Before applying the regime shift algorithm, we filled in larger gaps in the time series (e.q. significantly greater than 7 to 14 days between the mooring deployments) with data from other years. For all figures in this study containing RSI analysis,

we used data from the closest following year. This was done as the hydrographic properties are rapidly evolving in time, so conditions only a few years later would yield unrealistic results. Nonetheless, we tested the sensitivity to the choice of year by filling in the gaps with data from all other years as well. The results were mostly robust to the choice of year (not shown), thus confirming the occurrences and the timings of the regime shifts. Shorter gaps, such as between re-deployments of the moorings, were linearly interpolated over.





## 2.5 Limitations and uncertainties

There are some inherent uncertainties associated with producing long-term time series in the deep ocean, where gradients often are very small. This is particularly true for conductivity sensors (from which we derive $S_A$), which have a initial accuracy of around $\pm0.003$ g kg$^{-1}$, and are generally prone for drifts. For our time series of salinity, we observe typical ranges between 35.08 to 35.095 g kg$^{-1}$ (see next section). This means that the initial accuracy of $\pm$ 0.003 g kg$^{-1}$ represents 20% of the total variability, and the uncertainty is likely to increase as conductivity sensors drift. Additionally, we were not always able to correct individual deployments with calibration data from CTDs. To avoid clearly non-physical offsets, these deployments were manually adjusted to have the same mean as the rest of the time series, however this might act to remove or even introduce new signals. Nonetheless, as we will demonstrate, the general agreement between both F11 and HG-FEVI suggests that the salinity data is mostly correct, when considering the entire time span.

## 3 Results

### 3.1 Hydrographic changes between 1980-2023

We compile CTD data since the 1980's until as recently as 2023, to analyse the large scale hydrographic changes in the three areas of interest in this paper: Fram Strait, the Greenland Sea, and the Eurasian Basin (see Fig. 1 for boundaries of these areas).

Since the 1980s, we see a clear warming signal in the Greenland Sea from around -1.3 °C to -0.85 °C in the 2020's, and a salinity increase from $\sim$ 35.05 g kg$^{-1}$ to 35.085 g kg$^{-1}$ (Fig. 2a). In Fram Strait, which is influenced by both GSDW and EBDW, it is a bit more complex (Fig. 2b). Temperatures vary between -1.2 °C and -0.95 °C in the 1980's and overall warm to $\sim$ -0.85 °C by the 2020's. Salinities vary between 35.05 g kg$^{-1}$ and 35.1 g kg$^{-1}$ in the 1980's to $\sim$ 35.09 g kg$^{-1}$ by the 2020's. During the same time, the Eurasian Basin has warmed from $\sim$ -0.95 °C to -0.85 °C (Fig. 2c). There is also an indication of a slight salinity decrease in the Eurasian Basin, from around 35.098 g kg$^{-1}$ to 35.095 g kg$^{-1}$, although it should be noted that this is similar to the accuracy of 0.003 g kg$^{-1}$ of the salinity sensors. Overall, this has amounted to a density decrease of about 0.03-0.04 kg m$^{-3}$ in the deep Greenland Sea, and 0.02 kg m$^{-3}$ in the deep Eurasian Basin. This means that GSDW is still fresher, but has similar or higher temperatures than EBDW since $\sim$ 2015 (Fig. 2d). We also note that the waters in Fram Strait fall in between the mixing lines of EBDW and GSDW (Fig. 2a-c) , and can therefore be explained as a mixture of both end-members, and are overall dominated by the warming and salinification of GSDW.

Year-round, near-continuous time series from the moorings F11 and HG-FEVI in western and eastern Fram Strait, respectively, confirm the picture of an overall warming signal in the deep layers (Fig. 3a). For both moorings we see a warming from $\sim$ -0.9 °C to -0.85 °C (Fig. 3a). As the temperatures of GSDW and EBDW converge (see Fig. 2), we also observe reduced variability from $\sim$ 2016 and forward. At F11, we also observe the emergence of a seasonality from $\sim$ 2016 and forward, with higher temperatures in summer, and lower temperatures in winter. Time series of salinity are harder to interpret, due to the reduced accuracy of conductivity sensors compared to temperature sensors. Nonetheless, both time series oscillate between a saltier end-member and a fresher one, indicating the presence of EBDW and GSDW. Both F11 and HG-FEVI appear to be





**Figure 2.** Depth-averaged (2400-2600 m) $\Theta$-$S_A$ properties for regions a) Greenland Sea, b) Fram Strait, and c) Eurasian Basin. Grey dots show data for all basins, while coloured dots show basin-specific data and are coloured by year. Dots without black edges show CTD data, and dots with black edges show Argo data. Thin black lines show potential density anomaly referenced to 2500 dbar. d) Same $\Theta$-data as Fig. 2a-c, but plotted as a time series for Greenland Sea (blue), Fram Strait (black), and Eurasian Basin (red). The boundaries of the regions are defined in Sec. 2.1, and shown in Fig. 1. Data sources are listed in Sec. 2.1.





characterised by the more saline EBDW, with intermittent pulses of GSDW. For F11, we also observe reduced variability in salinity over time, possibly highlighting the observed salinification of GSDW apparent in the hydrographic profile data (Fig. 2).

From the hydrographic profile data it is clear that the still fresher GSDW has warmed to similar temperatures as EBDW, with individual measurements even showing higher temperatures for GSDW (Fig. 2). This is difficult to prove from sparse hydrographic data alone, due to the inherently large spatial and temporal variability when examining large regions as the Greenland Sea or the Eurasian Basin. To investigate changes, and timing thereof, in the deep ocean properties in Fram Strait, we cross correlated the normalised daily time series of $\Theta$ and $S_A$ (see Eq. 1) of the moorings F11 and HG-FEVI (Fig. 4). Both

time series exhibit similar patterns with an overall shift from a period with positively correlated properties (warm/salty and cold/fresh) up to 2014 to anti-correlated properties (warm/fresh and cold/salty) after 2017. The shift to predominantly anti-correlated properties occurs in late 2017 for F11 and in 2018 for HG-FEVI, respectively. Also, these shifts are not immediate, and we find a period of transition from $\sim 2013/2014$ to 2017, where the mean during that period is only weakly positively correlated.

**3.2    Relationship between water masses and ocean velocities**

Time series of velocity show that the mean flow at F11 in western Fram Strait is characterised by southward velocities, i.e. the deep extension of the East Greenland Current (Fig. 5a, blue colours). Additionally there is a large seasonality, with strong southward velocities in winter, and weak southward or northward velocities in summer. The zonal component for F11, is comparatively very small, but has a net westward flow (Fig. 5, blue colours). For HG-FEVI in eastern Fram Strait, the meridional

component instead shows a northward flow throughout the time series, i.e. the deep expression of the West Spitsbergen Current (Fig. 5a, red colours). The zonal component also shows a mean westward flow (Fig. 5b, red colours). A seasonal component is also observed for HG-FEVI, with stronger velocities towards the northwest in winter, and weaker velocities in summer. The seasonality with stronger currents along the deep bathymetry in winter and weaker in summer are resulting from the wind-driven gyre circulation in the Nordic Seas (Isachsen et al., 2003).

In order to see whether the flow field advects the water masses, or in contrast the density gradient between the water masses drives the flow, we plot Hovmöller-diagrams of cross-correlations between deep (near-bottom) meridional/zonal velocities and normalised $\Theta$ for F11 and HG-FEVI (Figs. 6 and 7, respectively). We then expect a band of stronger correlations centered around any lag. Note that due to the low precision of the salinity sensor, we have to limit this analysis to the temperature.

    F11 shows a general change from negative to positive correlations between temperature and velocities (Fig. 6). Furthermore,

we observe no significant change in velocities at F11 (Fig. 5), which are predominantly southward and weakly westward. The change in correlation therefore confirms our previous results: as GSDW (to the south) becomes warmer than EBDW (to the north), the southward flows switches from bringing relatively warm waters to bringing relatively cold ones over time (see Figs. 2, 4). Similarly, the westward flow is likely associated with bringing EBDW over to western Fram Strait, as there is a strong recirculation in Fram Strait (e.g. de Steur et al., 2014). Additionally, from 2015 onward, we again observe the emergence of

the seasonal band seen in Fig. 4, where there is a clear shift between positive and negative correlations. von Appen et al. (2015)





**Figure 3.** Time series of a) Conservative Temperature Θ and b) Absolute Salinity $S_A$ for moorings F11 (blue colours) and HG-FEVI (red colours) at approximately 2500 m depth. Small dots show individual measurements, and the thick lines show the 3-month moving average. We have zoomed in on the y-axes, as in 2013, a salty and warm plume from Storfjorden passed through HG-FEVI (von Appen et al., 2015), with a maximum temperature of -0.77° C and a salinity of 35.102 g kg$^{-1}$.

have shown that the velocity field advected the water masses with a lag of 1 day. We, however, do not detect any obvious lag dependency. At times we observe strong correlations for velocity leading temperature, and at times the opposite.



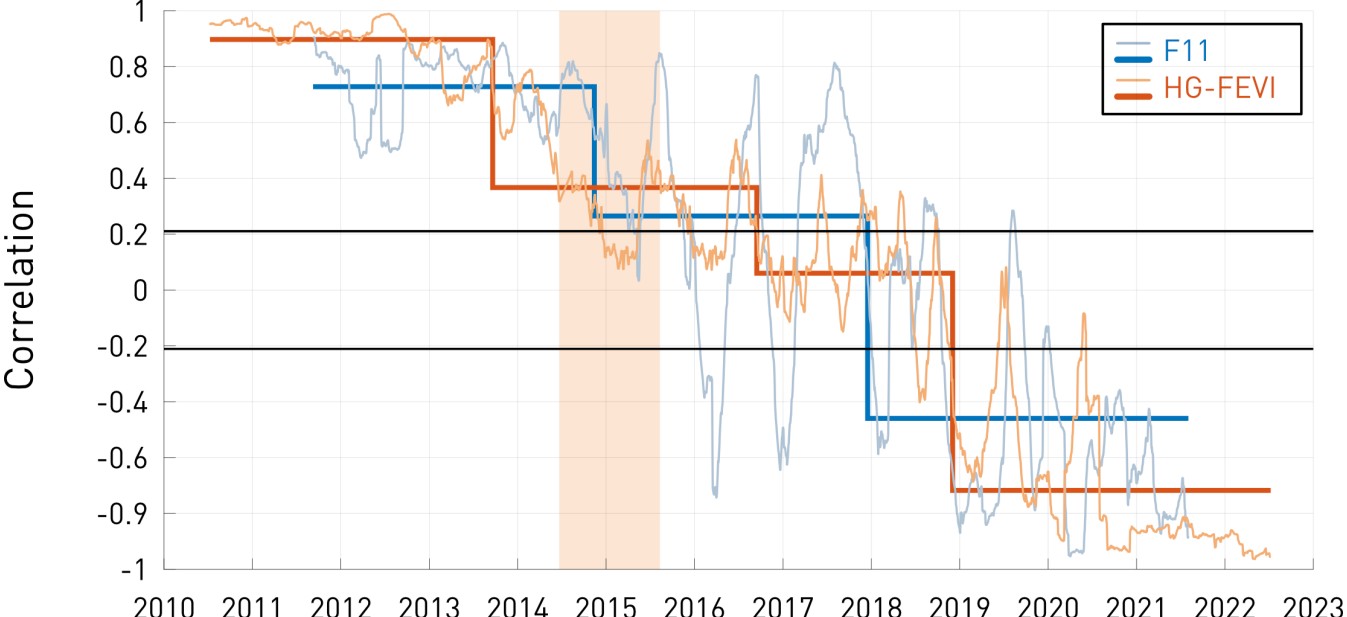

**Figure 4.** Time series of cross correlated variables. Thin lines show cross correlated normalised Conservative Temperature Θ and normalised Absolute Salinity $S_A$ for moorings F11 (blue colours) and HG-FEVI (red colours). The light orange box highlights the period with missing data for HG-FEVI, which was filled in with data from the following year. Black lines show 95% confidence levels; data within these levels are not statistically significant. See Sec. 2.3 for how time series of cross correlation were generated. Thick coloured lines show the mean for a certain period, as defined in Sec. 2.4.

At HG-FEVI, meridional velocities change from being positively to negatively correlated with the temperature (Fig. 7a), while the zonal velocities change from negatively to positively (Fig. 7b). Again, there is no significant change in velocities (Fig.

5), which are predominantly northward and westward. This means that like at F11, the flow switches from bringing relatively warm waters to bringing relatively cold ones. This can be explained by the fact that HG-FEVI is in fact EBDW-dominated (von Appen et al., 2015), and EBDW becomes relatively colder. Similar to F11 (Fig. 6), we also observe the emergence of a seasonality, however it should be noted that the signal is less pronounced than at F11. Like at F11, there is also no clear indication of a lag-dependency.

The emergence of a strong seasonal cycle in temperature could be indicative of the processes driving, or at least impacting the deep water mass changes. Therefore, we briefly investigate it further. The seasonal cycle in deep ocean velocities observed at both moorings, with stronger velocities in winter and weaker in summer (Fig. 5), indicates a strong connection to upper ocean flows, which in Fram Strait are known to have a strong seasonality (e.g. Beszczynska-Möller et al., 2012; de Steur et al., 2014). When investigating cross-sill advection, von Appen et al. (2015) found a strong correlation between upper ocean flows in the mesoscale band (3-30 days) and deep ocean velocities, suggesting that eddies barotropically force cross-sill advection in

the deep. Therefore, following von Appen et al. (2016), we calculate eddy kinetic energy (EKE) as:





**Figure 5.** Time series of a) meridional velocity component v and b) zonal velocity component u for moorings F11 (blue colours) and HG-FEVI (red colours) at approximately 2500 m depth. Small dots show individual measurements, and the thick lines show the 3-month moving average.

$$EKE = \sqrt{u^{*2} + v^{*2}} \qquad (2)$$



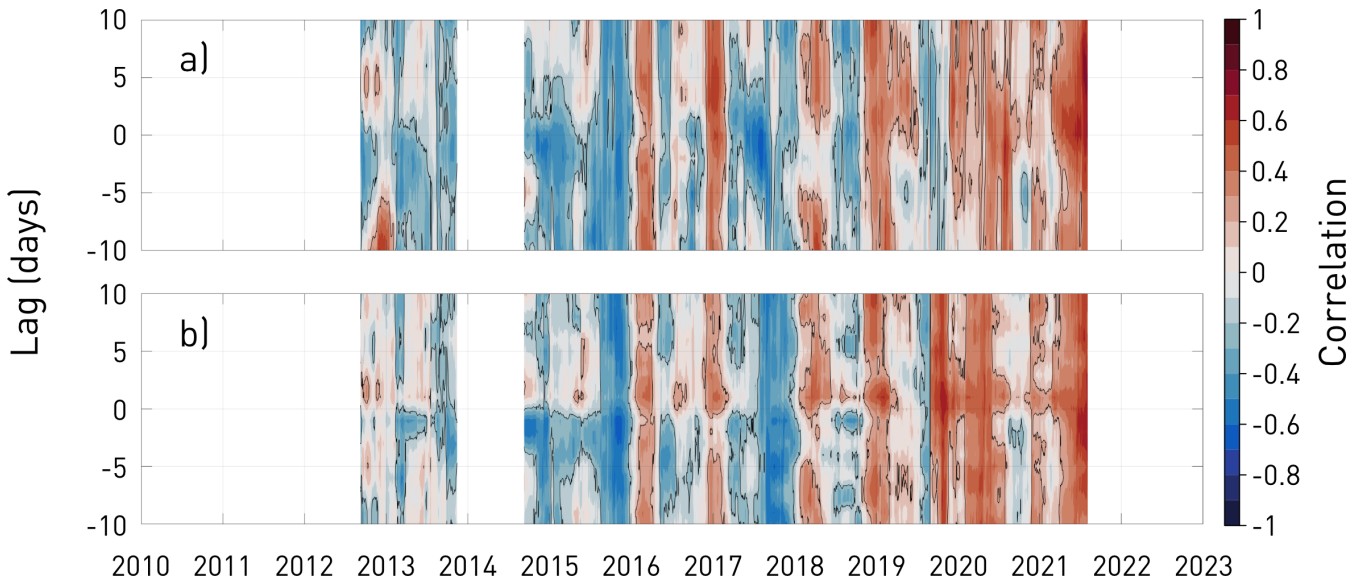

**Figure 6.** Hovmöller diagram of cross correlation between a) meridional velocity component V and normalised Conservative Temperature Θ and b) zonal velocity component U and normalised Conservative Temperature Θ for mooring F11. See Sec. 2.3 for how time series of cross correlations were produced. Red colours indicate a positive correlation, and blue colours a negative correlation. Negative lags indicate that velocity leads temperature, and positive lags indicate that temperature leads velocity. Black contours highlight 95% confidence levels.

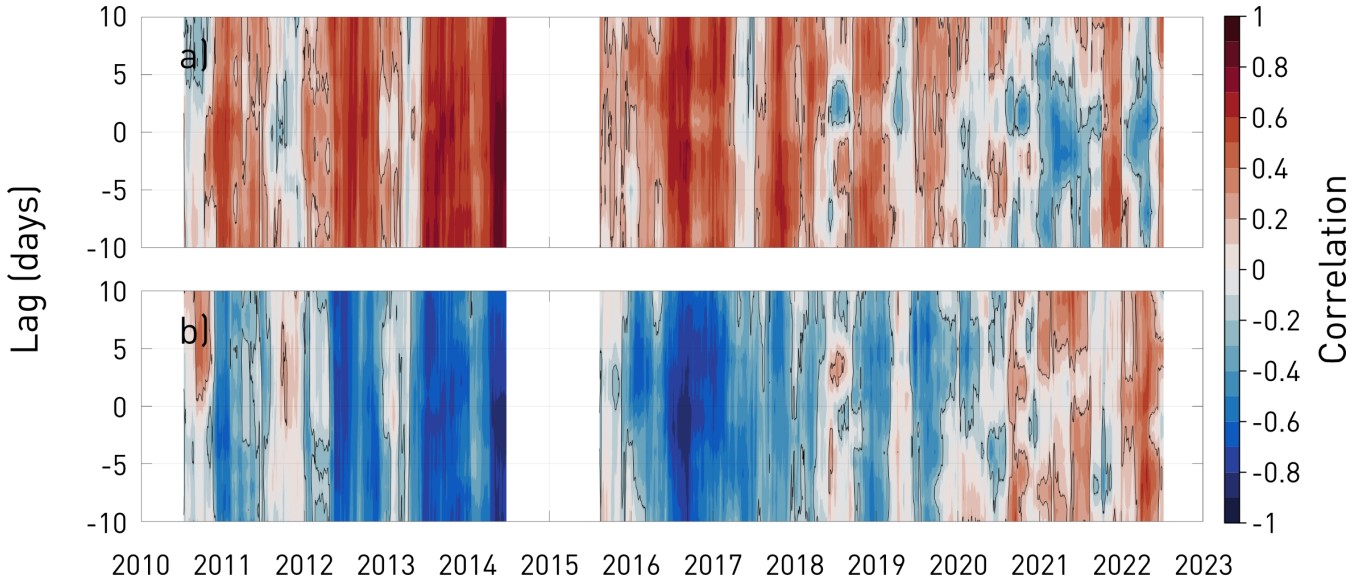

**Figure 7.** Same as Fig. 6, but for mooring HG-FEVI.





where u$^*$ and v$^*$ are the zonal and meridional velocity components, bandpassed between 3-30 days. For both moorings we indeed observe a clear seasonal cycle, with high EKE in March-April and low EKE in September (Fig. 8).

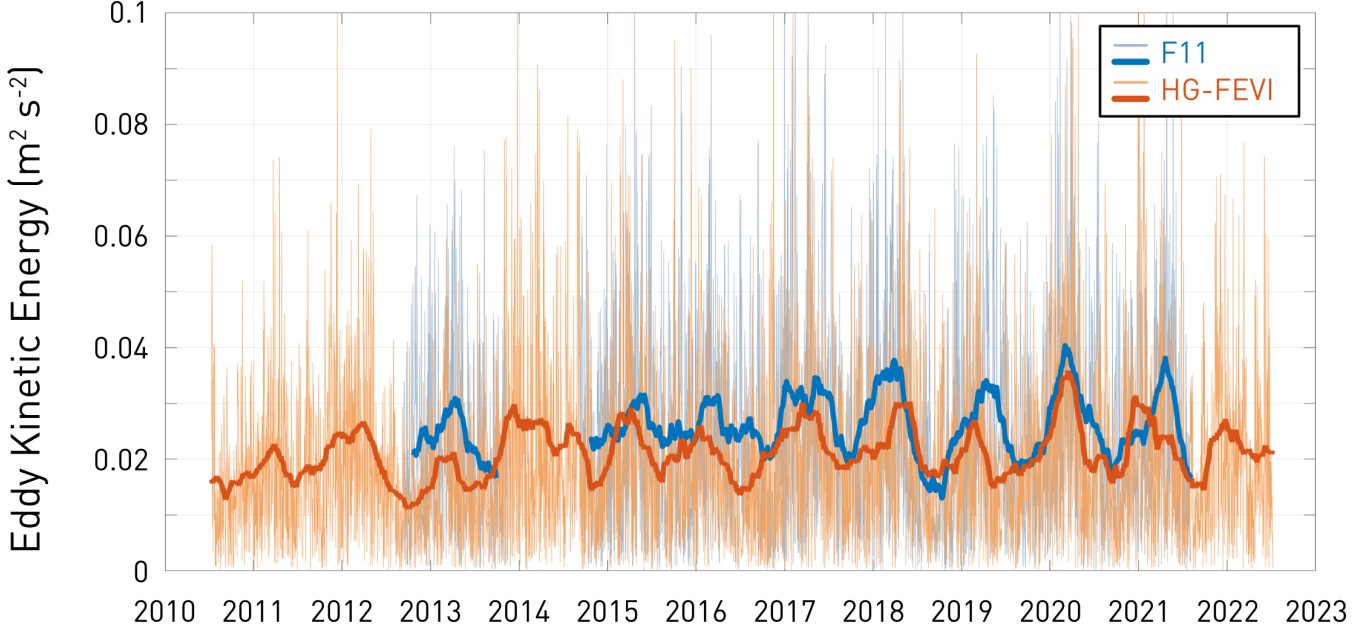

**Figure 8.** Time series of eddy kinetic energy (EKE) for moorings F11 (blue) and HG-FEVI (red). EKE calculated as $0.5 * \sqrt{u^{*2} + v^{*2}}$, where u$^*$ and v$^*$ are the zonal and meridional velocity components, respectively, bandpassed between 2-30 days. Thick lines show the 3-month moving average.

In summary, the temperature of the deep waters of the Greenland Sea and Eurasian Basin converged in the mid-2010s, a result we observe both in the CTD data and in the Fram Strait moorings. The salinities, however, are not changing as rapidly, and the magnitude and direction of the flow in Fram Strait are unchanged. Additionally, we observe the emergence of a seasonality in temperature. We will now discuss the possible drivers for these changes, and potential consequences.

## 4 Discussion

**4.1 Evolution of hydrographic properties**

Compiling hydrographic profile data since the 1980's, we have shown that GSDW has warmed to similar temperatures as EBDW in $\sim$ 2015, and likely already has become warmer (Fig. 2). This is difficult to prove from profile data alone, due to the large variability associated sampling across large regions. Nonetheless, using high-resolution mooring data, these results are confirmed by two moorings from the western and eastern Fram Strait, which clearly show that normalised $\Theta$ and S$_A$ have

turned from positively correlated (warm and salty/cold and fresh) to negatively correlated (warm and fresh/cold and salty) since late 2017/2018 (Fig. 4). This is also shown when comparing velocities to temperature at F11 (Fig. 6a), where there is a clear



shift from anti-correlation (the mostly southward flow used to bring comparatively warm Eurasian Basin waters) to positive correlation (the still-southward flow now brings comparatively cold EBDW). This means that the Greenland Sea is now a heat source for the Arctic Ocean at these depths. It naturally follows that we could expect the temperature of the Eurasian Basin to increase, as GSDW no longer provides a heat sink for EBDW. However, EBDW has been shown to mainly be composed of dense shelf waters from the Barents and Kara Seas (e.g. Bauch et al., 1995). The heat source of GSDW thus likely plays a second-order role to shelf sea properties and dynamics, as well as the warming of the intermediate waters (Lauvset et al., 2018), which the dense shelf waters entrain (e.g. Bauch et al., 1995; Rudels, 1986).

If GSDW was simply replaced by EBDW alone, the water masses would have converged to the same properties. However, in the hydrographic profile salinity data, it shows GSDW is still fresher than EBDW (Fig. 2). Since GSDW has warmed above the temperature of EBDW, it follows that it requires an additional heat source. One such possible source could be inflows of Norwegian Sea Deep Water (NSDW) through the Jan Mayen Channel (JMCh). Originally, NSDW was comprised of a mix of EBDW and GSDW, flowing cyclonically along the margins of the Greenland Sea and out through the JMCh (e.g. Swift and Koltermann, 1988). However, it has been concluded that since the 2000's, NSDW has evolved alongside GSDW properties, or been warmer and fresher than GSDW (von Appen et al., 2015), thus falling outside of the mixing line between GSDW and EBDW. This is also consistent with the observed reversals in the direction of the flow in the JMCh (Østerhus and Gammelsrød, 1999; Wang et al., 2021). However, it remains uncertain how long these reversal events last, ranging from several months (Østerhus and Gammelsrød, 1999), to highly intermittent events lasting only a few days (Pellichero et al., 2023). Thus, at present, it is unclear how large the contribution of NSDW to the temperature increases of GSDW is.

Another possible source could be convection in the Greenland Sea. Since the cessation of the deepest convection in the 1980's (e.g. Bönisch et al., 1997), the main product of the wintertime convection in the Greenland Sea has been the Greenland Sea Arctic Intermediate Water (GSAIW; Budeus et al., 1998; Latarius and Quadfasel, 2010; Jeansson et al., 2017). While originally a very cold and fresh water mass, formed with a large influence of fresh polar waters (e.g. Rudels et al., 2002), GSAIW is now formed by cooling of warm and salty Atlantic Water (e.g. Jeansson et al., 2017). Overall, this has resulted in a strong warming and salinification of the intermediate water column (Lauvset et al., 2018). The increased presence of salty AW in the Greenland Sea has led to a decreased stratification in the upper water column (e.g. Brakstad et al., 2019; Bashmachnikov et al., 2021). Consequently, it has been shown that the declining convection trend observed after the 1980's has been reversed, and convection in the Greenland Sea increased during the 2000's (Lauvset et al., 2018; Brakstad et al., 2019; Bashmachnikov et al., 2021). However, since 2014 it has been shown that both the convection depth and area has been reduced by approximately 50%, with a mean convection depth of ∼500 m (Abot et al., 2023). Therefore, it is unlikely that local convection in the Greenland Sea contributes directly to the warming observed at 2500 m depth by mixing down warmer waters. However, the possibility remains that GSAIW is replacing GSDW. There has been evidence for an upwelling cell at the JMCh was draining the deepest waters of the Greenland Sea into the Norwegian Sea (Somavilla, 2019). This would act to replace the deeper waters in the Greenland Sea with the above-lying intermediate waters.





## 4.2 Observed seasonality

At F11, we observe a clear emergence of seasonality in temperature from 2015 (Fig. 3a, blue colours), with warmer temperatures in summer, and colder temperatures in winter. Velocity data at F11 shows generally strong southward flows in winter and weak southward flows or northward flows in summer (Fig. 5). This indicates that the seasonal signal observed at F11 can be explained by advection of EBDW south in winter and GSDW north in summer.

At HG-FEVI, we observe at times indications of a seasonality in temperature and salinity, although the signal is in general obscured by what appears to be interannual variability (Fig. 3, red colours). This is generally consistent with previous studies where it has been shown that EBDW was the dominant signal at HG-FEVI, compared to the rest of the mooring line at 78.8° N, where the presence of EBDW and GSDW was found to be more evenly distributed (von Appen et al., 2015). Similar to F11 however, we generally find stronger velocities in winter and weaker velocities in summer (Fig. 5).

von Appen et al. (2015) found a strong correlation between velocities in the upper ocean in the mesoscale band (3-30 days) and in the deep. This suggests that surface eddies with an equivalent barotropic component force deep mesoscale ocean flows. Following von Appen et al. (2016) we computed EKE and found a clear seasonality, with higher values in winter and lower values in summer. Overall, this implies that cross-sill advection in the deep mainly occurs in winter when EKE is high. For F11, where we found that EBDW was mainly present during winter, this must be related with southward cross-sill transport.

## 5 Conclusions

In this study we provide an overview of the large scale hydrographic changes in the deep Fram Strait and its source regions, the Greenland Sea and the Eurasian Basin, by first compiling hydrographic profile data from 1980 to 2023 (Fig. 2). We find a strong warming trend in the Greenland Sea of $\sim$ 0.4-0.5° C since the 1980's. This warming trend is roughly an order of magnitude larger than in the rest of the World Ocean at the same depth level. During the same period, we find a modest temperature increase of $\sim$ 0.1° C in the Eurasian Basin. Additionally we find a strong salinification of GSDW and a weak freshening of EBDW. Using high-resolution mooring data in Fram Strait, we time GSDW becoming warmer than EBDW in late 2017/2018. Additionally, we observe the emergence of a strong seasonality in temperature at the western mooring, with an increased EBDW presence in winter and an increased GSDW presence in summer, caused by advection. Our results show that we may expect EBDW temperatures to rise faster in the future, as GSDW no longer acts as a heat sink but rather a heat source for the deep Arctic. We can however not estimate the consequences on e.g. the endemic ecosystem or on sea level rise as the deep ocean in general, and the deep Arctic in particular, are tragically under-observed. Our findings demonstrate once again the crucial role that long-term, high resolution time series play for our understanding on the changing Arctic, even at great depths.

*Data availability.* The mooring data from NPI mooring F11 are freely available in its original form via https://doi.org/10.21334/npolar. 2021.c4d80b64. The mooring data for AWI mooring HG-FEVI between 2010-2015 are available in a processed form with reduced precision https://doi.org/10.1594/PANGAEA.845616, https://doi.org/10.1594/PANGAEA.845618, https://doi.org/10.1594/PANGAEA.845620,



https://doi.org/10.1594/PANGAEA.845622, https://doi.org/10.1594/PANGAEA.861858. The raw data for AWI mooring HG-FEVI between 2015-2022 are freely available via https://doi.org/10.1594/PANGAEA.870848, https://doi.org/10.1594/PANGAEA.904538, https://doi.org/10.1594/PANGAEA.904539, https://doi.org/10.1594/PANGAEA.942180, https://doi.org/10.1594/PANGAEA.946514, and https://doi.org/10.1594/PANGAEA.964045. Hydrographic profile data from The Unified Database for Arctic and Subarctic Hydrography (UDASH) are

320     freely available via https://doi.pangaea.de/10.1594/PANGAEA.872931. Hydrographic profile data from the World Ocean Database 2018 (WOD18) are freely available via https://www.ncei.noaa.gov/products/world-ocean-database. Hydrographic profile data from the Argo program are freely available via https://argo.ucsd.edu/. Hydrographic profile data from yearly zonal sections across Fram Strait conducted by NPI are freely available via https://doi.org/10.21334/npolar.2014.e3d4f892, https://doi.org/10.21334/npolar.2022.493ea7ad, https://doi.org/10.21334/npolar.2022.52ecdc98, https://doi.org/10.21334/npolar.2022.29c6e2c7, https://doi.org/10.21334/npolar.2022.44db5c55, https:

325     //doi.org/10.21334/npolar.2022.2c646c2e, https://doi.org/10.21334/npolar.2022.5066a075, https://doi.org/10.21334/npolar.2022.5df344c6, and https://doi.org/10.21334/npolar.2022.17b6bec5. Hydrographic profile data from cruises KVS2007 and KVS2008 are freely available via https://doi.org/10.21343/7jqb-5930 and https://doi.org/10.21343/btym-vh89, respectively. Hydrographic profile data from cruise JCR2018 are freely available via https://doi.org/10.5285/84988765-5fc2-5bba-e053-6c86abc05d53. Hydrographic profile data from the Multidisciplinary drifting Observatory for the Study of Arctic Climate (MOSAiC) expedition are freely available via https://doi.org/10.1594/PANGAEA.

330     959963. Hydrographic profile data from I/B Oden expedition Synoptic Arctic Survey (SAS) is freely available via https://doi.org/10.1594/PANGAEA.951266. Hydrographic profile data from R/V Kronprins Haakon expedition AO22 is freely available via https://doi.org/10.21334/npolar.2022.d1e609e2. The gridded bathymetry from the International Bathymetric Chart of the Arctic Ocean (IBCAO) is freely available through https://www.gebco.net/data_and_products/gridded_bathymetry_data/arctic_ocean/. The land mask from A Global Self-consistent Hierarchical High-resolution Geography Database (GSSHG) is freely available via https://www.soest.hawaii.edu/pwessel/gshhg/.

335     *Author contributions.* All authors contributed to designing the study; M.H. and L.d.S. provided the mooring data; S.K. and C.H. further calibrated the mooring data; S.K. conducted most of the analysis, under the supervision of C.H. and L.d.S. All authors contributed to the manuscript.

*Competing interests.* The authors declare no conflict of interest.

*Acknowledgements.* This work was funded via Vetenskapsrådet grant 2018-03859 awarded to Céline Heuzé.



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
