# Peer review of "Continued warming of deep waters in Fram Strait"

_EGUsphere, 2024_

## Author Comment (AC1)

**Reviewer 1:**

We thank Reviewer 1 for the time they spent reading and commenting on the manuscript. Your contribution has been noted in the acknowledgement section:

"We thank the two anonymous reviewers and the editor Agnieszka Beszczynska-Möller for their comments, which greatly improved this manuscript."

In the abstract, you just mention "deep ocean", specify that you talk about 2500m, XXm above the bottom, and at sill depth between the 2 basins, also L301. It would be useful to sharpen exactly what you mean by "deep ocean" in this context and which layers (above and below) you do not consider in your analysis. It is fine that you don't consider them, but it would be good not to pretend that you cover the whole ocean below the mixed layer. e.g. Langehaug&Falck, unlike you, consider "intermediate and deep waters".

We thank the reviewer for pointing this out. We have now rephrased L3 in the abstract, as well as L301 in the conclusions to be more precise and better reflect that we looked at changes close to the sill depth of Fram Strait.

L16 all these references deal with the Greenland Sea which often is considered Nordic Seas rather than Arctic Ocean. At least clarify what your definition of Arctic Ocean is then.

We have now defined the Arctic Ocean, as the Nordic Seas and the central Arctic Ocean. This is also consistent with the definition by the International Hydrographic Office.

L72 The north-south extent of the Fram Strait box is actually quite a bit larger than around the sill across the strait. Maybe comment on the meridional extent of the sill separating the basins.

We thank the reviewer for their comment. After L72 we have now added: "For the definition of Fram Strait, we chose a larger meridional extent compared to the sill. We did this as we expect that the mixed waters from Fram Strait are not just limited to the sill but might be seen north and south of the strait."

L76 to the exact choice of

Done.

L82 typo: -0.7 --> -0.8degC

Changed

L91 moorings F10...

Done.

L107 conductivity which translates to XX for salinity

We assume you mean L104 and have added "which translates to 0.003 g kg$^{-1}$ for Absolute Salinity."

Tab 1 4.03degE -- 4.333degE. I don't think the range of the location in the zonal direction was that large. The mooring was always close to 4.33degE = 4deg20minE

Almost all deployments were deployed close to 4.33 °E, however the 2013-14 deployment was deployed at 4.03 °E. We have made no changes to the text.

L117 Are those in the bottom boundary layer? By comparison to the treatment of the temperature/salinity data, the info on velocity data is comparably brief. Do the links to data sets that you give for temperature/salinity also contain the velocity data?

Yes, the current meters were deployed at the same depth as the MicroCATs, and the datafiles we link also contain the velocity records. We have now clarified that in the text.

Fig 2 what is the -1degC 35.05 excursion of GSDW in 2000?

Might note that (deep!) Argo is only 6 profiles in GSDW in one year, thus it does not really contribute to your conclusions.

You could show the locations of the profiles in Fig. 1 as dots in the basins.

We thank the reviewer for pointing this interesting feature out. We are not sure of underlying dynamics; however, it appears that the freshening observed in the early 2000's is related to a freshwater anomaly further up in the water column (Fig. R1-1 of this document) and a deepening of the isopycnals. This deepening of isopycnals is clearly visible in Fig. 5 of Brakstad et al., 2019 (https://doi.org/10.1175/JPO-D-17-0273.1). We have now added a sentence at L175: "although in 2003-2005 we observe a strong freshening, likely caused by deepening of fresher intermediate waters".

Indeed, we only have 6 profiles from Deep Argo, however they extend our hydrographic profile time series until 2023 in the Greenland Sea and therefore provide the most recent data we use in this study. We have nonetheless added a sentence at L66: "Although we note that since Deep Argo floats were only recently deployed in the Greenland Sea, we were only able to acquire 6 profiles."

We have now added the locations of the profiles as dots in Fig. 1a.

[Figure]

Fig. R1-1. Depth-averaged salinity (1400-1600 m) in the Greenland Sea.

L204 Fig 5b

Fixed

Fig 6 caption Clarify whether the direction of the velocity impacts whether it is +1 or -1. This might be especially useful when in Fig 7 caption you just state that it is the same.

We have now added a sentence in the figure caption stating: "Note that the sign of the velocity itself is near-constant throughout the time series".

Eqn 2: EKE is 1/2 times what you show in eqn 2. This is correct in Fig 8 caption, though it is not clear why you need the info in the figure caption and the main text.

We thank the reviewer for pointing this out. We have corrected the error in eq. 2 and removed the equation from the caption in Fig. 8. We would like to point out that the error in eq. 2 was only in the text and has not impacted the results.

L248 associated with

Fixed

L256 "thus likely plays" maybe replace by "may play". The causal relationship why one should be more important in the net than the other is not clear to me.

Done.

L260 However… grammar!

We agree that the sentence was poorly written at best. We have now replaced it with "However, the hydrographic profile data shows that GSDW is still fresher than EBDW".

L269 increase

Fixed

L284 What about vertical mixing unrelated to convection?

While there is some vertical mixing unrelated to convection, especially close to boundaries where tide-topographic interactions can occur, this is likely rather small. As explained in Somavilla, 2019 (https://doi.org/10.1029/2018JC014249), high rates of vertical mixing would lead to a homogenisation of the water column, which is not observed in observations (Fig. 9 in Somavilla (2019)). Instead, a continuous decrease in temperature is observed rather than a bottom mixed layer. It therefore seems more likely that the deep waters of the Greenland Sea are being upwelled out of Greenland Sea and being replaced by e.g. EBDW and GSAIW.

We have now added in the text at L289: "Previously, high rates of vertical mixing have been reported in the deep Greenland Sea, which would quickly homogenise any gradients in the deep ocean (Budeus and Ronski, 2009). However, this is not observed in recent observations (Somavilla, 2019)."

L311 "tragically" not sure that is the best word. Maybe tone down a bit.

Replaced with "still"

L312 understanding the

Changed

L464 data set from 2010 contains data until 2015?

Yes, the DOI was created in 2010, but the dataset was updated until 2015.

---

## Author Comment (AC2)

**Reviewer 2:**

First, we thank Reviewer 2 for taking the time to read the manuscript and comment on it. Your contribution has been noted in the acknowledgement section:

"We thank the two anonymous reviewers and the editor Agnieszka Beszczynska-Möller for their comments, which greatly improved this manuscript."

L3: Define "deep ocean"

After a comment that Reviewer 1 also had, we have specified here and replaced "deep ocean" with "close to the sill depth of 2500 m".

L5: Suggest replacing "upstream basins" with "surrounding basins" or similar, as the basins can also be "downstream" in terms of the flow.

Excellent point, we have replaced it now with "adjacent basins".

L22: Rapid warming of GSDW or less GSDW in the mixture? Related to my comment on the water mass definition (L76-77) below.

The fraction of "pure" GSDW has decreased, which has caused the overall warming. We replaced here "rapid warming of GSDW" with "rapid warming of the deep Greenland Sea".

L31: Do you mean mixing in Fram Strait or exchange across the strait? I think exchange is a better word here.

Both are valid, e.g. von Appen et al., 2015 (https://doi.org/10.1016/j.dsr.2015.06.003) showed that strong mixing of these deep waters occurs at the sill. We have nonetheless changed that sentence to: "Fram Strait is thus a key region for exchange between the Arctic-derived EBDW, formed by shelf-slope convection and entrainment of intermediate waters, and waters from the Nordic Seas, such as GSDW, which was formed by open-ocean convection in the Greenland Sea gyre (e.g. Rudels, 2012; Langehaug and Falck, 2012)."

L36: "… GSDW now being a lighter …": I'm missing a reference here. Is this based on Somavilla et al., 2013? Then the word "now" is not appropriate. Rather specify when the transition happened.

We thank the reviewer for pointing this out. It is based on von Appen et al., 2015 (https://doi.org/10.1016/j.dsr.2015.06.003). We have now added the reference and specified when the transition occurred.

L51: "knowledge of …, since 2012/2014, is lacking.": why do you include two different years?

We wrote this way because von Appen et al., 2015 (https://doi.org/10.1016/j.dsr.2015.06.003) bases most of their study on mooring data that ends in 2012, and one mooring that ends in 2014. However, we acknowledge that the writing was confusing and have replaced "2012/2014" with "2014".

L53-61: I found the start of this paragraph (L53-57) a bit difficult to follow because the main goal and focus of the paper are not mentioned in the introduction yet. It would be good to clarify this in the beginning of this paragraph (instead of later in L58-59).

We thank the reviewer for their suggestion, we now lead the paragraph by stating the overarching objective of this paper.

L73: Is there a typo in the east-west extension of the GS box in the text? It says 12W-12E (same as the FS box), but on the map it doesn't look like the GS box goes to 12E (rather 0E, which makes more sense too, in terms of excluding the NS).

This was indeed a typo, and it was supposed to say "0 °". We have now corrected it. We also note that this was just a typo in the text, and it has not affected the results in any way.

L76: exact choice "of" depth levels

Fixed

L76-77: I agree that it is suitable to use a property-independent definition to look at deep water changes in specific regions. However, I think it would be good to clarify that the deep waters you look at contain (are mixtures of) water masses with different origin. GSDW is often referred to in the literature as a class of water that was formed in the Greenland Sea prior to the 1980s. Your GSDW definition likely contains an increasing amount of other water masses such as EBDW and GSAIW. You do discuss this later on in the discussion but it would be good to include one or two sentences here to avoid confusion.

We thank the reviewer for raising a good point. We have now clarified: "Here we note that GSDW was a cold and fresh water mass formed by deep convection before the 1980's, and that the deep waters of the Greenland Sea since then contain an increasing amount of other waters, such as EBDW (von Appen et al., 2015). For the sake of simplicity, we nonetheless refer to the deep waters of the Greenland Sea as GSDW."

L80-83: Please be consistent with the number of decimals you present for the salinity values (and temperature values). This is also the case in other parts of the manuscript (e.g., section 2.5 and the first paragraph of section 3.1).

We have now changed the values to be consistent in number of decimals when comparing numbers directly against each other.

L81: … which "corresponds to a" change of … in Practical Salinity …    (same for temp.)

Fixed

L95-98: Does this mean that you exclude deployments during the 2010-2022 period or are the deployments with Aanderaa RCMs before 2010 (which is why you start in 2010)? This is not clear. Also, the Aanderaa RCMs are dataloggers/platforms with point current meters and the option to install different types/versions of temperature sensors (etc.) with different specs. Which temperature sensor (product number) has the accuracy you refer to of +/-0.05degC?

We thank the reviewer for pointing this out, this was indeed not clear in the text. The deployments with the RCMs are before 2010, or 2011 (for F11). We have now made this clear. The RCMs were installed with thermistors of type Fenwall GB32JM19, which now also has been clarified in the text.

Table 1: "Not all data is always available during the deployments." I think this needs some elaboration. Is it because of gaps in the deployments, sensor failures, or because you excluded deployments with RCMs or other reasons?

They were due to sensor failures; we have now clarified this in the text of Table 1.

L121: "… compare our results to previous studies, …": which results? I'm missing one introductory sentence about why you are doing the cross-correlation analysis (like the first sentence you have in section 2.4, L139-140).

This was indeed vague, we have now clarified: "In order to distinguish between the two water masses EBDW and GSDW, we followed a similar procedure to von Appen et al., 2015, where we normalised the daily-averaged temperature and salinity data from the moorings."

L124: Remove one "the" in "We define the the upper and lower bounds…".

Fixed

L124-126: "We define the upper and lower bounds by …, to get estimates of the upper and lower bounds of water mass properties.": The last part of this sentence is not needed/is a repeat of the start (ie. remove the part after "to get estimates ….").

Done

L152-153: "The results were mostly robust to the choice of year (not shown), ..": It is difficult to know what "mostly robust" means. Could you be more quantitative or give an example?

We have replaced "mostly robust" with "robust". The results did not change except for one instance when the missing 2014/15 for HG-FEVI was replaced with data from 2021/22, which would represent a very anomalous year considering how much the temperature has changed over time.

L157: …, which have "an" initial accuracy …

Fixed

L175-176 and Fig 2: What does the density change correspond to in terms of potential density anomaly referenced to the surface? (particularly for the DW in the Greenland Sea).

It is somewhat difficult to assess due to large variability in the 1980's, however when referenced to the surface, the potential density of GSDW has remained relatively constant (Fig R2-1). However, since the halt of deep convection in the 80's, diapycnal mixing has been limited, and we see no reason to refer the potential density to the surface, since changes mainly occur isopycnically with EBDW. We have therefore not made any changes to the text.

[Figure]

Fig R2-1. Time series of depth-averaged (2400-2600 m) potential density referenced to the surface.

L176-177: It is not clear from the data in fig.2d that the GSDW temp is higher than EBDW after ~2015. That is, I would just state that they have converged here. Evidence of larger GSDW temp comes primarily from the mooring data in Fram Strait and the cross-correlation analyses later.

Good point, we have changed the phrasing here to state that the temperatures have converged.

L217: either "flow switches" or "flows switch"

Changed

Fig. 4 caption: … as defined "by the regime shift analysis" in Sec. 2.4.

Done

L226: Remove "in fact" before "EBDW-dominated".

Removed

L230-231: If I understand it correctly it is primarily the seasonal cycle in velocity combined with the different rates of warming of GSDW and EBDW that leads to the stronger seasonality in temperature. What about seasonality in temperature in the upstream water masses?

Yes exactly, this is our hypothesis. It is possible that there is some level of seasonality in the upstream water masses, although we are not sure what could dynamically force such a seasonality. In any case, at present we have no way of assessing seasonality in the upstream water masses due to a lack of data. We have nonetheless added a sentence on how we cannot rule out a seasonality in the upstream water masses in the discussion at L301.

Equation 2: multiply by 1/2

Done. We note that this was a typo in the text only, and has not affected the results in anyway.

L242: ... direction of the "deep" flow in Fram Strait ...

Added

L248: ... associated "with" sampling ...

Added

L282: "However, the possibility remains that GSAIW is replacing GSDW." Could you plot typical properties of GSAIW to check this? What about geothermal heating? Could this be another source of the increased temperature at 2500m?

After both yours and R1s comments regarding this we have added text to better explain: "However, the possibility remains that GSAIW is replacing GSDW. Previously, high rates of vertical mixing has been reported in the deep Greenland Sea, which would quickly homogenise any gradients in the deep ocean and form bottom mixed layers (Budeus and Ronski, 2009). However, this is not observed in recent observations (Somavilla, 2019). Instead there has been evidence for an upwelling cell at the JMCh, draining the deepest waters of the Greenland Sea into the Norwegian Sea (Somavilla, 2019). This would act to replace the deeper waters in the Greenland Sea with the above-lying intermediate waters, and is consistent with the deepening of isopycnals observed in the Greenland Sea (Brakstad et al., 2019; Somavilla, 2019)"

Considering the strong isopycnal deepening observed in e.g. Fig 5 of Brakstad et al., 2019 (https://doi.org/10.1175/JPO-D-17-0273.1), we are confident that GSAIW is replacing the deep waters. This is also consistent with results from Somavilla et al., 2013

(https://doi.org/10.1002/grl.50775). If you look at Table 2., it is also showing a stronger warming from above rather than from below, suggesting that GSAIW is an extra heat source.

L312: … play for understanding the changing Arctic. Remove "our" and "on" before and after "understanding", respectively.

Done